Studying turn performance, trunk control, and mobility in acute stroke subjects: a cross-sectional study

Vasyani Mahima 1
Nayak Akshatha 1 akshatha.nd@manipal.edu
http://orcid.org/0000-0003-4937-7442 Kumar K. Vijaya 1
Misri Zulkifli 2
http://orcid.org/0009-0002-6508-6557 Choezom Pema 1
http://orcid.org/0009-0009-1007-6109 Mascarenhas Rinita 3
http://orcid.org/0000-0002-2876-4227 Tedla Jaya Shanker 4
http://orcid.org/0000-0002-2686-0397 Natarajan Srikant 5
1 Department of Physiotherapy, Kasturba Medical College, Mangalore, Manipal Academy of Higher Education , Manipal, Karnataka , India
2 Department of Neurology, Kasturba Medical College, Mangalore, Manipal Academy of Higher Education , Manipal, Karnataka , India
3 Department of Neurology, Christian Medical College , Ludhiana, Punjab , India
4 Program of Physical Therapy, Department of Medical Rehabilitation Sciences, College of Applied Medical Sciences, King Khalid University , Abha , Saudi Arabia
5 Department of Oral Pathology and Microbiology, Manipal College of Dental Sciences, Mangalore, Manipal Academy of Higher Education , Manipal, Karnataka , India
van den Hoek Daniel
Electronic publication date: 2024 Dec 3
Publication date: 2024
Volume: 12
Electronic Location ID: e18501
Received 2024 May 24; Accepted 2024 Oct 18
Copyright: © 2024 Vasyani et al.
Copyright year: 2024
Copyright holder: Vasyani et al.
License: This is an open access article distributed under the terms of the Creative Commons Attribution License, which permits unrestricted use, distribution, reproduction and adaptation in any medium and for any purpose provided that it is properly attributed. For attribution, the original author(s), title, publication source (PeerJ) and either DOI or URL of the article must be cited.
License URL: https://creativecommons.org/licenses/by/4.0/

Keywords: Stroke, Turning, Trunk performance, Trunk control, Mobility

Funding: Deanship of Research and Graduate studies at King Khalid University through Large Research Project RGP2/216/45 This work was supported by the Deanship of Research and Graduate studies at King Khalid University through Large Research Project under grant number RGP2/216/45. The funders had no role in study design, data collection and analysis, decision to publish, or preparation of the manuscript.

==============================
Background

Stroke leads to various impairments like motor deficits, impaired trunk control and restricted mobility. However, rehabilitation professionals often underestimate the fundamental function of turning, which is essential for daily living activities like walking, cooking, or performing household chores. Impaired turning can be attributed to motor deficits post-stroke, resulting in restricted mobility and impaired trunk movement. Therefore, the present study aimed to determine the relationship between turn performance, trunk control, and mobility in stroke patients.

Materials and Methods

A total of 63 first-time supratentorial stroke (i.e., anterior circulation stroke) patients aged 18–90 years were recruited for the study. Turn performance was assessed by asking patients to walk for 10 feet comfortably, then take a 180° turn and return to the starting position. In addition, the duration and number of steps were recorded. Following this, the Trunk Impairment Scale (TIS) and Stroke Rehabilitation Assessment of Movement (STREAM) were used to assess trunk impairment and mobility, respectively. The group turn performance was analyzed using the Kruskal–Wallis test with a post hoc Mann–Whitney U test for between-group comparisons. The turn duration and turn steps were correlated with age, trunk control, and mobility using Spearman’s rank correlation. A regression analysis was performed to determine the association of turn performance with age, trunk control, and mobility among stroke patients.

Results

Thirty stroke patients had turning difficulty, and 33 did not. Hence, they were categorized into the turning difficulty (TD) and non-turning difficulty (NTD) groups. When correlated with turn duration and the number of steps taken by the stroke patients while turning, the STREAM and TIS scores revealed a significant negative correlation (p < 0.001). The subjects’ age showed a significant positive correlation with the turn duration and number of steps taken by stroke patients while turning (p < 0.001). A significant association was also found between turn performance and age and trunk control. However, there was no significant association between turn performance and mobility.

Conclusion

The observed associations highlight the complexity of turning ability and trunk control necessary to complete a turn safely. Additionally, with advancing age, turn performance and turning movement are compromised in stroke patients. This indicates that turning difficulty is more pronounced in older individuals with stroke.

Introduction

Globally, stroke is the third leading cause of death and disability combined (Feigin et al., 2021). Stroke causes motor impairments, which result in the restriction of activities of daily living (ADLs), functional impairments and restriction of participation in society. Evidence suggests that most stroke patients showed impairments in their functional and basic mobility (Poomalai, Prabhakar & Sirala Jagadesh, 2023).

Stroke patients stagger and exhibit unsteadiness, which limits the execution of smooth functional movements (Rohrer et al., 2002; Faria et al., 2016). Fifty percent of community-dwelling stroke survivors fall, and a large proportion of these falls occur while turning (Hyndman, Ashburn & Stack, 2002). This could be attributed to motor dysfunction caused by the stroke, which impairs the temporal and spatial coordination of the head, trunk, and pelvis (Lamontagne et al., 2005). Turning constitutes a rudimentary aspect of ADLs, accounting for more than 40% of turning functions in complex activities (Glaister et al., 2007). Turning facilitates directional changes during ambulation, targeted navigation, and overcoming obstacles; thus, its safety and efficacy are paramount. This task demands intricate coordination of muscle groups, perceptual acuity, vestibular processing, and proprioceptive feedback (Taylor, Dabnichki & Strike, 2005). Additionally, turning necessitates precise postural adjustments, axial realignment, and gait parameter adaptation, underscoring its intricate biomechanical nature in human locomotion (Xu, Carlton & Rosengren, 2004). Turning also requires a stable trunk and balance (Sandin & Smith, 1990), making it challenging for stroke patients, as demonstrated by the greater number of steps and time required to complete a turn at each of the angles tested previously (Lam & Luttmann, 2009). However, poor turn performance is not exclusive to post-stroke individuals; it is also observed in the aging population (Thigpen et al., 2000). Nevertheless, among post-stroke individuals, poor turn performance is due to impaired trunk control (Verheyden et al., 2006, 2007; Karthikbabu et al., 2012; Liang et al., 2021). Trunk control and truncal impairment are predictors of the comprehensive function of ADLs among the stroke population (Smith, Barber & Stinear, 2017; Ishiwatari et al., 2021).

Previous literature has evaluated turn performance and turning ability in stroke patients (Lam & Luttmann, 2009; Manaf et al., 2012). Additionally, studies have been performed on trunk control, trunk function impairment, functional and basic mobility, and motor recovery after stroke (Langhorne, Coupar & Pollock, 2009; Karthikbabu et al., 2012). However, there is a dearth of literature that correlates turn performance with trunk control and mobility post-stroke. Therefore, the present study aims to determine the relationship between turn performance, trunk control, and mobility in acute stroke patients.

Materials and Methods

Study design

In this cross-sectional study, a total of 63 acute stroke participants were included to evaluate turn performance and determine its association with trunk control and mobility. Ethical approval for the study was obtained from the Institutional Ethics Committee (IEC) of Kasturba Medical College, Manipal Academy of Higher Education, Mangalore, India. Upon approval (IECKMCMLR-01/2023/12), this study was registered in the Clinical Trials Registry—India (CTRI/2023/11/059768).

Study participants

Sixty-three participants diagnosed with a first episode of supratentorial stroke (i.e., anterior circulation stroke) and admitted to a tertiary care hospital in Mangalore from January 2023 to February 2024 were included in the study (Fig. 1). Participants who were clinically stable, aged 18–90 years, scored ≥26 on the Montreal Cognitive Assessment (MoCA), and could walk independently without walking aids or orthoses were included. Participants were excluded if they had any neurological conditions other than stroke, visual or perceptual deficits, musculoskeletal or cardiovascular conditions, and comorbid disabilities that could affect the assessment. All the participants were informed regarding the assessment and provided written informed consent before recruitment.

Figure 1 Participant flowchart.

Outcome variables

Trunk impairment scale

The Trunk Impairment Scale (TIS) assesses trunk control and motor impairment post-stroke. The static subscale examines the ability of the subject to sustain a seated position with the feet supported and maintain it with legs both passively and actively crossed. The dynamic subscale includes items related to trunk lateral flexion and unilateral lifting of the hip. To assess trunk coordination, the subject is asked to rotate the upper or lower part of his/her trunk six times. For each item, a 2-, 3-, or 4-point ordinal scale is used. The overall score of the TIS ranges from 0 for the worst possible performance to 23 for the best possible performance. Intraclass correlations (ICCs) for the total scores of various subscales were between 0.85 and 0.99. The test-retest and inter-observer reliability for the TIS total score (ICC) was 0.96 and 0.99, respectively. Furthermore, Cronbach alpha coefficients reflecting internal consistency ranged between 0.65 and 0.89 (Verheyden et al., 2004).

Stroke rehabilitation assessment of movement

The Stroke Rehabilitation Assessment of Movement (STREAM) is a commonly utilized assessment tool for evaluating the restoration of voluntary movement and basic mobility after a stroke. It is employed to gauge a patient’s coordination, functional mobility, and range of motion. The STREAM consists of 30 items that are equally distributed among three subscales: upper and lower leg movements and mobility items. The weighted kappa statistics for inter-rater agreement on individual item scores ranged between 0.55 and 0.94, while the ICC for the total score was 0.96, indicating high inter-rater reliability (Wang et al., 2002; Hsueh et al., 2006).

VibraTilt application

VibraTilt is a dual-purpose application that functions as both a gyroscope and an accelerometer utilizing the portability and ease of access of smartphones and tablets. It offers additional features such as calibration, resetting, and color customization for the accelerometer, alongside measurements and graphing capabilities for the gyroscope. Users can specify the x-, y-, and z-axis range (measured in rad/s) for measurement using the gyroscope function. It is a free app available for both Android and iOS that unlocks these extra features for the popular smartphone. The primary distinguishing feature of VibraTilt is its function of measuring shakes within a pre-determined duration, sensitivity and threshold. This versatility makes VibraTilt the first of its kind, suitable for both scientific research and recreational use (Ng, Nguyen & Gan, 2016).

Procedure

Demographic data (age, gender, and dominant side) and stroke-related details (affected side, post-stroke duration, and site of lesion) were recorded. Trunk control was assessed using the TIS and mobility was assessed using the STREAM. The participants were thoroughly explained the study procedure and given two trials, followed by a 10-min break. Later, the final reading was taken, wherein the subject’s turning ability and performance were assessed by asking them to walk for 10 feet at a comfortable speed, take a 180° turn in whichever direction felt most comfortable for them, and then return to the starting position (Fig. 2A). A smartphone was strapped around the patient’s body below the chest level. The gyroscope limit in rad/s and duration were set in the VibraTilt app. Then, the “Start” button was pressed to initiate the measurement. To end the measurement, the “Stop” button was pressed. The turn time and rate of rotation were noted in the graph panel (Ng, Nguyen & Gan, 2016) (Fig. 2B). The number of steps was assessed through the video clips recorded using a camera mounted on a tripod focusing on the turn (Fig. 2A). The turn test was performed with the shoes off on level ground for all participants during both trials and the final assessment. Based on turning performance, the stroke patients were divided into turning difficulty (TD) and non-turning difficulty groups (NTD). Subjects who needed more than 3 s or five steps to perform a 180° turn were considered to have poor turning performance and included in the TD group, and the remaining participants were considered to have better turning performance and included in the NTD group based on the criteria mentioned in a previous study (Thigpen et al., 2000). To avoid assessment bias, the subject’s turning performance, trunk control, and mobility were tested in a random order.

Figure 2 Assessment of Turn performance.

Data analysis

Data entry and statistical analyses were performed using the Statistical Package for Social Sciences (IBM SPSS Statistics for Windows, Version 28.0, Armonk, NY). The Shapiro–Wilk test was used to test the normality of data. The turn duration and number of steps among the groups (TD and NTD) were analyzed using the Kruskal–Wallis test with a post hoc Mann–Whitney U test for between-group comparisons. The dependent variables (turn duration and turn steps) were correlated with age, TIS, and STREAM using Spearman’s rank correlation. A binary stepwise forward logistic regression was used to determine the association of turn performance with age, trunk control, and mobility among stroke patients. p < 0.05 was considered significant.

Results

A total of 63 individuals were recruited for the study, of whom 30 were in the TD group and 33 were in the NTD group. The mean age of the TD group participants was 65.53 years and that of the NTD group participants was 59.33 years. The age range in the TD and NTD groups was 45 to 83 years and 43 to 73 years, respectively. The subjects in the TD group were older than those in the NTD group, and this difference was statistically significant (p = 0.02). The mean post-stroke duration of the TD group participants was 5.6 days and that of the NTD group participants was 4.97 days. The post-stroke duration range of the TD and NTD groups was 3 to 9 days and 3 to 8 days, respectively, and the difference was not significant (p = 0.54). All subjects included in this study were right-hand dominant, including 36 women and 27 men, and the distribution of subjects based on gender was not significant (p = 0.64). When the side of weakness among the stroke patients was compared across the groups, the NTD group had approximately an equal distribution of subjects with right-sided and left-sided weakness. However, among the TD group, most of the subjects had right-sided weakness, and this difference was statistically significant (p < 0.001) (Table 1). The mean score of STREAM in the TD and NTD groups was 59.33 and 61.94, respectively. The mean score of TIS in the TD and NTD groups was 18.23 and 20.79, respectively (Table 2).

Table 1 Descriptive data of stroke subjects.

Variables		n	Group	t value/chi square	p value	
			TD	NTD			
			n (mean ± SD)	n (mean ± SD)			
Age (Years)	63	30 (65.53 ± 11.14)	33 (59.33 ± 9.05)	2.434	0.02*	
Post-stroke duration (Days)	63	30 (5.60 ± 1.89)	33 (4.97 ± 1.72)	5.05	0.54	
			n (%)	n (%)	Chi square		
Gender	Female	36	19 (63.3)	17 (51.5)	0.90	0.64	
Male	27	11 (36.7)	16 (48.5)	
Affected side	Left	25	9 (30.0)	16 (48.5)	130.49	<0.001*	
Right	38	21 (70.0)	17 (51.5)	
Notes:

TD, Turning Difficulty; NTD, Non-Turning Difficulty; SD, Standard Deviation.

* Significant, p < 0.05.

Table 2 Mean scores in STREAM and TIS of included stroke patients.

Variables	Group	
TD (mean ± SD)	NTD (mean ± SD)	
STREAM	59.33 ± 3.22	61.94 ± 2.44	
TIS	18.23 ± 2.98	20.79 ± 1.19	
Note:

TD, Turning Difficulty; NTD, Non-Turning Difficulty; SD, Standard Deviation.

Turn performance

Stroke patients who needed more than 3 s and five steps to accomplish a 180° walking turn were classified as having poor turning performance. Based on the abovementioned criteria, stroke patients’ turning performance was classified as TD or NTD. A total of 30 stroke patients were included in the TD group, and the remaining 33 subjects were included in the NTD group. The time taken to accomplish a 180° turn varied between the groups, with a mean duration of 4.8 s in the TD group and 2.85 s in the NTD group. The time required by the TD group to perform a 180° turn was statistically significantly longer (p < 0.001) than that of the NTD group. The number of steps taken while taking a 180° turn also differed between the groups, with a mean value of 9.1 steps in the TD group and 4.64 steps in the NTD group. The TD group took a statistically significantly greater number of steps (p < 0.001) than the NTD group (Table 3, Fig. 3).

Table 3 Turn performance of stroke subjects.

	TD (N = 33) Mean ± SD	NTD (N = 30) Mean ± SD	NTD vs. TD Mean difference (p value)	
Turn (seconds)	4.8 ± 1.19	2.85 ± 0.36	−1.95 (<0.001)*	
Turn (steps)	9.1 ± 2.44	4.64 ± 0.7	−4.46 (<0.001)*	
Notes:

TD, Turning Difficulty; NTD, Non-Turning Difficulty; SD, Standard Deviation.

* Significant, p < 0.05.

Figure 3 Turn performance of stroke subjects.

Correlation analysis

The STREAM (r = −0.44) and TIS (r = −0.50) scores showed a significant negative correlation with turn duration (p < 0.001). Moreover, the STREAM (r = −0.45) and TIS (r = −0.54) scores showed a significant negative correlation with the number of steps taken by the participants while turning (p < 0.001). Stroke patients with higher impairment of trunk control and lower mobility required more time and a greater number of steps while taking 180° turns. The subject’s age showed a significant positive correlation with turn duration (r = 0.44) and the number of steps (r = 0.50) taken while turning (p < 0.001). Older subjects required more time and a greater number of steps to take a 180° turn, whereas younger participants took a lower duration and a smaller number of steps (Table 4, Fig. 4).

Table 4 Correlation of turn performance with age, TIS, and STREAM.

	Turn (seconds)	Turn (steps)	
Age	r = 0.44	r = 0.50	
p < 0.001*	p < 0.001*	
TIS	r = −0.50	r = −0.54	
p < 0.001*	p < 0.001*	
STREAM	r = −0.44	r = −0.45	
p < 0.001*	p < 0.001*	
Notes:

TD, Turning Difficulty; NTD, Non-Turning Difficulty; TIS, Trunk Impairment Scale; STREAM, Stroke Rehabilitation Assessment of Movement; SD, Standard Deviation.

* Significant, p < 0.05.

Figure 4 Correlation of turn steps and turn duration with TIS, STREAM, and Age.

Regression analysis

A binary stepwise forward logistic regression was used to predict the association of the dependent variables turn (steps) and turn (seconds) with age, trunk control, and mobility. The results showed a significant association of turn duration with trunk control (β = −0.313, p = 0.011) and age (β = 0.373, p < 0.001). Similarly, there was a significant association of turn steps with trunk control (β = −0.356, p = 0.002) and age (β = 0.426, p < 0.001). However, the majority of the subjects significantly showed more weakness on the right side; the affected side of participants (left and right) did not show an association with turn performance. Additionally, mobility skills did not show an association with turn time (β = −0.220, p = 0.073) and turn steps (β = −0.206, p = 0.070) (Table 5 and Fig. 5).

Table 5 Binary stepwise forward logistic regression of turn duration and turn steps with age, TIS, affected side, and STREAM.

		Unstandardized Coefficients	Standardized coefficients	t	p value	95.0% confidence interval for B	
		B	Std. Error	Beta			Lower bound	Upper bound	
Turn duration	(Constant)	9.226	2.862		3.224	0.002	3.497	14.956	
Age	0.046	0.012	0.373	3.723	0.000*	0.021	0.071	
TIS	−0.159	0.061	−0.313	−2.615	0.011*	−0.281	−0.037	
Affected side	0.240	0.267	0.091	0.900	0.372	−0.294	0.774	
STREAM	−0.092	0.051	−0.220	−1.824	0.073	−0.193	0.009	
Turn steps	(Constant)	17.880	5.782		3.092	0.003	6.307	29.454	
Age	0.116	0.025	0.426	4.608	0.000*	0.065	0.166	
TIS	−0.396	0.123	−0.356	−3.221	0.002*	−0.643	−0.150	
Affected side	0.549	0.539	0.095	1.019	0.312	−0.530	1.628	
STREAM	−0.188	0.102	−0.206	−1.846	0.070	−0.393	0.016	
Notes:

TD, Turning Difficulty; NTD, Non-Turning Difficulty; TIS, Trunk Impairment Scale; STREAM, Stroke Rehabilitation Assessment of Movement.

* Significant, p < 0.05.

Figure 5 Regression plots for turn steps and turn duration.

Discussion

This study’s objective was to assess turn performance in stroke patients based on the number of steps and time taken to accomplish a 180° turn and correlate these parameters with the following factors: age, trunk control, and mobility. The results revealed poor turn performance in acute stroke patients with and without turning difficulty. Stroke patients in the present study required a longer duration and more steps to perform a 180° turn compared with those in a previous study of the chronic stroke population (Chen et al., 2021). Additional steps during turning are believed to indicate instability and diminished coordination, suggesting a greater risk of falls among acute stroke patients. However, we hypothesize that turn performance might be enhanced in chronic stroke patients as they may adopt a strategy to minimize steps given their familiarity with their impairments compared to acute stroke patients (Thigpen et al., 2000; Dite & Temple, 2002; Fuller, Adkin & Vallis, 2007).

Among the correlated variables, age demonstrated a strong positive correlation with both the number of turning steps and the time taken to complete a 180° turn. In the TD group, which was found to be older, the number of steps and time taken to perform the 180° turn were greater, whereas younger stroke patients typically required fewer steps and less time to complete the turn. This aligns with previous research indicating that turning behavior and performance are compromised in stroke survivors (Manaf et al., 2012). Older people were previously reported to have altered balance and reduced muscle strength in the lower limbs (Osoba et al., 2019; Bullo et al., 2020), which was related to turning difficulty (Thigpen et al., 2000).

Slowing down the speed of the turn and allowing more time for turning can be a helpful tactic for older individuals to execute turns safely and effectively. Slowing down during a turn may aid in widening the base of support from side to side, thus enhancing stability. Older individuals often adopt a careful walking pattern characterized by slower steps, shorter strides, and prolonged periods of having both feet on the ground simultaneously (Imms & Edholm, 1981). These adjustments aim to reduce the risk of losing balance and cope with balancing difficulties while walking (Shkuratova, Morris & Huxham, 2004; Osoba et al., 2019). Additionally, as healthy individuals age, there is a decline in turning performance and speed accompanied by an increase in turning duration (Weston et al., 2024). Swanson & Fling (2020) found that the decrease in turn velocity and the prolongation of turn duration imply a more cautious and simplified turning strategy associated with advanced age. Additionally, the risk of falls increases with age. Impaired balance and falls are more prevalent in older adults (Chauhan, 2013). This could explain the poor turning performance in older adults with stroke. In the present study, unlike age, the affected side of stroke patients did not appear to have any significant impact on their turning ability.

The trunk serves as the central support structure for the body (Cholewicki, Panjabi & Khachatryan, 1997). Trunk control is the ability to maintain a stable trunk while performing ADLs (Davies, 1990) and is impaired among post-stroke subjects (Verheyden et al., 2006, 2007; Karthikbabu et al., 2012). Stable trunk control is essential for maintaining balance in various postures and movements (Karthikbabu et al., 2012; Ishiwatari et al., 2021). The sitting balance (static or dynamic) is altered in post-stroke individuals primarily due to compromised trunk control (Verheyden et al., 2006, 2007); this, in turn, makes individuals more susceptible to falls (Ryerson et al., 2008). Trunk impairment is one of the predominant factors affecting turning ability, which is required for ADLs, when sitting and standing (Verheyden et al., 2006, 2007). This diminished trunk control complicates turning activities, as indicated by the strong negative correlation between turn steps, turn duration, and TIS scores in the current study. Individuals with lower TIS scores took more steps and required a longer time to complete a 180° turn, highlighting that reduced muscle strength impairs trunk control and further impacts turn capacity. The current study’s findings indicate that stroke patients with higher levels of mobility require less time and fewer steps to complete a 180° turn. Conversely, those with lower levels of mobility require more time and steps for the same task. Impaired mobility enables post-stroke individuals to execute turns with alterations and deviations, leading to longer turn durations and a higher number of steps (Lam & Luttmann, 2009; Manaf et al., 2012). Carey et al. (2005) stated that stroke patients have impaired motor recovery, mobility and motor control. This could be the reason for reduced mobility post-stroke, and hence, impaired turn performance.

The Time to Walk Independently after STroke (TWIST) algorithm considers trunk control to be a predictor for post-stroke mobility (Smith, Barber & Stinear, 2017); however, the literature on trunk control in acute stroke is limited. A strength of our study is the assessment of both trunk control and turn performance in acute stroke patients. Future studies can incorporate turn performance as a predictor in predictive models for stroke. Furthermore, we used a smartphone-based accelerometer and gyroscope application (VibraTilt) to assess turn performance (turn duration and turn steps). The key features of the VibraTilt application are a user-friendly interface and adjustable threshold, sensitivity, and time limit for the number of shakes.

This study has some limitations, the first being the lack of standardization in the direction of the 180° turn among participants. Additionally, we did not evaluate the turn direction for both the left and right sides. Instead, participants were permitted to turn 180° in the direction they found most comfortable. Future research should focus on studying the turn performance in different turn directions and turning tasks and determining its relationship with mobility and trunk function. In the present study, only supratentorial stroke population were included. Hence, the findings cannot be generalized to the other stroke population. Furthermore, stroke patients in our study were not classified according to the site of the lesion. The lesion site may have impacted the individual’s turn performance, trunk function, and mobility.

Conclusions

The present study provides insights into the association of turn performance with trunk control and mobility. Among the stroke patients assessed, the TD group showed poor turn performance with more steps and a longer duration needed to perform a 180° turn. Individuals with better trunk control and lower levels of motor impairment had better turn performance. Advancing age was also found to have a detrimental effect on attaining optimal turn performance. Hence, post-stroke rehabilitation should focus on assessing and improving the turning ability of stroke patients in the early phase of recovery by improving trunk control, essentially leading to greater mobility outcomes.

Supplemental Information

Supplemental Information 1 STROBE Checklist.

Supplemental Information 2 Raw data.

Additional Information and Declarations

Competing Interests

Author Contributions

Human Ethics

Data Availability

The authors declare that they have no competing interests.

Mahima Vasyani conceived and designed the experiments, performed the experiments, prepared figures and/or tables, authored or reviewed drafts of the article, and approved the final draft.

Akshatha Nayak conceived and designed the experiments, authored or reviewed drafts of the article, and approved the final draft.

K. Vijaya Kumar conceived and designed the experiments, authored or reviewed drafts of the article, and approved the final draft.

Zulkifli Misri conceived and designed the experiments, authored or reviewed drafts of the article, and approved the final draft.

Pema Choezom performed the experiments, prepared figures and/or tables, and approved the final draft.

Rinita Mascarenhas performed the experiments, prepared figures and/or tables, and approved the final draft.

Jaya Shanker Tedla conceived and designed the experiments, authored or reviewed drafts of the article, and approved the final draft.

Srikant Natarajan performed the experiments, analyzed the data, authored or reviewed drafts of the article, and approved the final draft.

The following information was supplied relating to ethical approvals (i.e., approving body and any reference numbers):

Institutional Ethics Committee, Kasturba Medical College, Mangalore, granted approval to carry out the study within its Facilities (IECKMCMLR-01/2023/12).

The following information was supplied regarding data availability:

The raw data is available in the Supplemental File.

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
