# Peer review of "Studying turn performance, trunk control, and mobility in acute stroke subjects: a cross-sectional study"

_PeerJ, doi:10.7717/peerj.18501_

## Round 0.1 · original submission · Major Revisions

Thank you for your submission. The reviewers have highlighted the need for English language editing and greater detail in several sections of your manuscript.

Your introduction needs to be more concise and direct to the topic of stroke rather than meandering with other content. Further, the methods section lacks clarity and detail that would make the trial repeatable by another research group. Therefore, please provide details associated with the comments from reviewers.

Reviewer 1 ·

Basic reporting

The manuscript requires English editing, and more general editing for clarity and correctness or wording and statements in various sections.

The structure of the introduction also requires some revision. For example, stroke is first introduced mid way in the second paragraph, but then more fundamental information about stroke is only introduced in the third paragraph. My suggestion is to re-order this to introduce stroke, different impairments resulting from stroke, then how this impacts daily activities, and finally, turn performance.

Given the age range of participants, there introduction would also benefit from some further presentation of how ADL and turn performance are impacted by age, not just stroke.

Data figures would benefit from displaying individual participant data. Present ranges/SDs for other participant demographic data. Some figures of regression analysis would also be useful to further visualise the findings.

Experimental design

The experimental design is not unreasonable but lacks substantial information.
1. Section would benefit from a flow chart of recruitment. Were any potential participants excluded?
2. There is no real references the specific motor impairments for participants prior to being part of the study.
3. Could all participants provide consent? Did any have speech or intellectual impairment post-stroke as this is an important consideration.
4. It seems unlikely that this patient group had no comorbidities or other significant health issues.
5. More detailed information about the app, and data it collects is required.
6. Was only one turn trial conducted? If so this is a limitation. It would also make sense to ask participants to trial turning each way for a better profile of turn performance.
7. Description of TIS and STREAM tests seems to include results information. Or, is this reliability info that is already published about the test?
8. Was age different between the groups?
9. Figure implies turn test was done with shoes off. Please make sure all relevant details for study replication are provided in methods.
Fig 1b. Radian value seems to have reached a ceiling. Device or measurement error?

Validity of the findings

Suggest that some further information about methods, and additional transparency of data is needed to help determine the validity of the results.

Conclusions section: The authors state that stroke subjects with higher levels of motor recovery demonstrate better turn performance. I am not sure this is accurate given the proximity of the tests with the occurrence of the stroke. Perhaps this is more about those with less severe motor impairment resulting from the stroke in the first instance? The effects of rehabilitative interventions would need to be investigated separately. L302-304 also appears the imply the opposite to what the authors first state here.

Authors state that both turn direction were not assessed which limits the findings.

Lesion site not used for classification of participants. This ties in with my 'specific motor impairment' comment. In my opinion the authors should explore these data to provide a more robust assessment and interpretation of the results.

Additional comments

The manuscript requires substantial revisions in the areas noted above.

Check format of some in text references e.g., L304.

Reviewer 2 ·

Basic reporting

Need to improve sentence structure throughout and tables should have been included within results. Had difficulty comparing raw data to conclusions within paper.

Experimental design

Added comments to article

Validity of the findings

added comments to article

Additional comments

If recorded, all in the way the clients turned.

Annotated reviews are not available for download in order to protect the identity of reviewers who chose to remain anonymous.

---

## Round 0.2 · Minor Revisions

Thank you for modifying your manuscript. You have made substantial revisions that address the reviewer's comments.

Please address the final minor comments from Reviewer 1 in the form of modifications, explanations, or rebuttals.

Reviewer 1 ·

Basic reporting

Check manuscript for some word spacing issues and inconsistencies in reference list formatting.

Experimental design

There are still some obvious limitations - some of these have been addressed but some still need to be further clarified.

Validity of the findings

Be mindful of over generalising the findings.

Additional comments

Specific comments to address:

L26-29: Long sentence.
There are some word spacing errors throughout manuscript.
L93-94: The difference between motor and functional impairment is not clear here.
L97: All of them or just ones with motor impairment?
Would still like to see some reliability and validity statistics for the Vibratilt device. The statement and reference provided is not informative.
L843: The sample size is decent but not large. Additionally, this only accounts for specific types of stroke and so making generalisations is difficult.
I still think 1 trial is limited. Can the authors present the data from the practice trials as well? It would be good to know things such as: was there a learning effect, what was the variability of turn performance between trials etc.
I don’t understand the authors response in relation to figure 1b. This still means that there was a ceiling and hence if this is true it means that the data would haven been impacted. Further clarity is needed.
Figure 3 and 4 – consider some further formatting adjustments so that data is more centralised in the figures.

---

## Round 0.3 · accepted · Accept

Thank you for taking the time to modify your manuscript in line with reviewer feedback. You have appropriately addressed the reviewer comments and your manuscript is now suitable for publication.